# The Role of Past Suicidal Behavior on Current Suicidality: A Retrospective Study in the Israeli Military

**DOI:** 10.3390/ijerph18020649

**Published:** 2021-01-14

**Authors:** Leah Shelef, Jessica M Rabbany, Peter M Gutierrez, Ron Kedem, Ariel Ben Yehuda, J. John Mann, Assaf Yacobi

**Affiliations:** 1Department of Health and Well-Being, IDF’s Medical Corps, Israel Defense Forces, Ramat Gan 5262000, Israel; ariel.benyehuda@gmail.com; 2Arrowhead Regional Medical Center, Colton, CA 92324, USA; jessicarabbany@gmail.com; 3Rocky Mountain Mental Illness Research, Education and Clinical Center, Rocky Mountain Regional VA Medical Center, Aurora, CO 80045, USA; Peter.Gutierrez@va.gov; 4Department of Psychiatry, Anschutz Medical Campus, University of Colorado School of Medicine, Aurora, CO 80045, USA; 5Statistican, Medical Corps-Israel Defense Forces, Ramat Gan 5262000, Israel; ron.kedem56@gmail.com; 6Division of Molecular Imaging and Neuropathology, Department of Psychiatry, Columbia University, New York, NY 10032, USA; jjm@columbia.edu; 7New York State Psychiatric Institute, New York, NY 10032, USA; 8Beer Ya’akov-Ness Ziona Mental Health Medical Center, Beer Yaakov 70350, Israel; assafyacobi@gmail.com; 9Sackler Faculty of Medicine, Tel Aviv University, Tel Aviv 6997801, Israel

**Keywords:** suicide attempt, suicide plan, suicidal ideation, suicidal threats, military service

## Abstract

Past suicide attempts are a significant risk factor for future suicidality. Therefore, the present military-based study examined the past suicidal behavior of soldiers who recently made a severe suicide attempt. Our sample consisted of 65 active-duty soldiers (61.5% males), between the ages of 18 and 28 years old (M = 20.4, SD ± 1.3). The inclusion criterion was a recent severe suicide attempt, requiring at least a 24 h hospitalization. This sample was divided into two groups, according to previous suicidal behavior, namely whether their first suicide attempt was before or after enlistment (*n* = 25; 38.5% and *n* = 40; 61.5%, respectively). We then examined the lethality and intent of the recent event in regard to this division. Four measures were used to assess the subjects’ suicidal characteristics: the Columbia Suicide Severity Rating Scale, the Self-Harm Behavior Questionnaire, the Suicidal Behaviors Questionnaire-Revised, and the Beck Scale for Suicide Ideation. No significant difference in the severity of the suicide attempts (either actual or potential severity) were found between those who had suicide attempts before enlistment and those who had their first attempt in the service. As a matter of fact, most of the suicide attempts that occurred for the first time during military service had used a violent method (58.3%, *n* = 21). Finally, using multivariate analyses, we found that current thoughts and behavior, rather than past suicidality, was the strongest predictor for the lethality of suicide attempts.

## 1. Introduction

Suicide is the 10th leading cause of mortality in adults in the United States [1] and the foremost cause of death in 15–34 year-olds [2]. For every completed suicide there are an estimated 10–20 suicide attempts (SA) [1], with suicidal ideation (SI) being even more prevalent. Approximately 4% of the population in the U.S. experience suicide ideation [3]. The best predictor of a first lifetime SA is a recent onset of SI [4]. Previous SA serves as one of the strongest predictors of a severe SA or suicide completion [5]. Between 50–78% of individuals attempting suicide die on their first attempt [6]. Among persons who attempt suicide, 1.6% die by suicide within the next 12 months [7]. A violent and severe suicide attempt was found as a robust risk factor for future suicide [8,9].

In a large study of 84,850 adults, it was found that 33.6% of people with SI eventually form a suicide plan and 29.0% attempt suicide [10]. Thus, suicidal ideation plays a key role in initiating the suicidal sequence [4,11,12].

Soldiers with a history of multiple suicide attempts reported the most severe suicide ideation [13]. Most of the transitions from ideation to attempt amongst US soldiers occur within the first year of the onset of ideation [14,15]. Soldiers with higher military occupational stressors (e.g., combat units) experience more negative effects [16]. Among soldiers, chronic stressors (vs. acute stress) were more strongly correlated with severity of suicide ideation [13].

Moreover, in the US Army, suicide rates have continued to rise over the past 15 years [17]. Therefore, understanding and predicting this transition from suicidal ideation to an attempt is of critical importance not only for the US Army but for soldiers everywhere [18].

Previous studies in the Israeli Defense Force (IDF) found that most soldiers who die by suicide during their military service did not have any known history of suicidal behavior [19]. While the US Army excludes enlistment of applicants with prior history of psychiatric disorders [20], in the Israeli military, those who have a history of such difficulties may still be deemed able to serve. However, they are recruited under special conditions with regular monitoring. In the enlistment process, candidates are interviewed and assessed. Since military service in Israel is compulsory, candidates with emotional or behavioral disturbances referred to a military clinical mental health professional are classified in two broad categories of diagnosis. Adjustment difficulties are defined as a cluster of personality traits limiting functionality and adaptability in the military service context. Psychiatric diagnosis, based on International Classification of Diseases (ICD-10), requires an assessment of the impact of the classification on the soldier’s potential functioning see the reference [21,22].

There are several barriers to identifying and treating suicidality for soldiers in active duty, primarily the tendency of soldiers to avoid seeking help [23,24]. This reluctance is most likely due to both the sense of pride that the soldiers embrace and the fear of reporting depression and hopelessness, since that could jeopardize their potential for promotion [25,26]. Perceived stigma surrounding psychiatric disorders in the military is also a barrier to help-seeking [27,28]. The combination of these factors may explain why many individuals with a known history of past suicidal ideation or behavior never seek or receive psychological help in the military [14,29,30]. However, it may also be the result of an acute onset of suicidal thoughts and subsequent impulsive behavior, as previous studies by our group have shown [19]. Support for this assumption was also found in studies performed in the US military. Among men, psychiatric disorders, characterized by agitation and impulsiveness, predict the transition from suicide ideation to attempt [31]. 

The present study, which is based on a prior study by our group [32], examined soldiers who recently made a severe suicide attempt. The aim was to identify variables that might have utility in predicting the characteristics of suicidal behavior occurring during military service. A severe suicide attempt was defined as any suicide attempt which mandated at least a 24 h stay at the hospital. We focused on the relationship between the current lethality and intent of the recent suicide attempt with regards to past suicide behavior, specifically focusing on the timing relating to enlistment. 

While most studies examine those who attempt suicide as a homogeneous group relative to control participants, the current study investigated whether there are significant differences between individuals who make a severe suicide attempt, based on their past suicidal behavior. To achieve this aim, we divided the study subjects into two groups: the first included soldiers with a history of a suicide attempt that happened before enlistment (hereafter: “SA before enlistment”) and the second included soldiers whose first suicide attempt occurred after enlistment (hereafter: “first SA in service”). Using this division of the study group, it was possible to examine the relationship between past and current suicide attempts and then to assess the lethality and intent of the recent event in regard to past suicidal behavior.

## 2. Methods

### 2.1. Participants

The study sample consisted of 65 active-duty IDF soldiers (61.5% male), between 18 and 28 years old (M = 20.4, SD ± 1.3) who agreed to participate in this study, while serving their compulsory military service. Most (70.8%) attempted suicide within the first year of military service, with half of these individuals (50.8%) attempting suicide during their first 6 months in service (i.e., after recruitment). Almost half of those who attempted (44.6%, *n* = 29) were discharged from military service after the suicide attempt. Under military procedures and following regulations of chain of care, those who remained in military service continued to be treated and observed by military mental health professionals throughout their service. 

Following the suicide attempt, all of the participants were examined by military psychiatrists. Table 1 presents the main diagnoses known before and after this examination. Anxiety and depression were the most common diagnoses in both groups, and a total of 61.5% of the subjects had suffered from these symptoms. Personality disorders (mainly cluster B disorders) were diagnosed in 33.9% of cases. There were no significant differences between the groups (SA before enlistment vs. first SA in service groups). For further information about the study group diagnoses, see the reference [32].

As mentioned above, the study sample was divided into two groups according to previous suicidal behavior before enlistment (soldiers with a history of a suicide attempt that happened before enlistment) or after enlistment (soldiers whose first suicide attempt occurred after enlistment), as self-reported in questionnaires and based on information gathered during the hospitalization and interviews after the suicide attempt. It should be noted that 8 (12.9%) subjects denied attempting to die by suicide. However, despite formal classification as non-suicidal self-injury (NSSI), these subjects were included in the study since their acts were severe enough to mandate a 24 h stay at a general hospital and endanger their lives. In addition, in many cases the history of suicidal behavior was unknown to military authorities, since they did not declare it before, and was only made known during the study investigation. The details of the study population are presented in Table 1.

Soldiers whose most recent suicide attempt was during their military service but whose first suicide attempt(s) was prior to induction into the military (*n* = 24; 38.1%) were categorized in the SA before enlistment group; soldiers whose first suicide attempt was after induction into the military (*n* = 40; 61.5%) were categorized in the First SA in service group. The majority of subjects in this group (*n* = 21; 56.8%) did not report prior suicide attempts, besides the most recent one. The rest were subjects who made at least one more suicide attempt after joining the army (in these cases, they did not report these attempts to military authorities). Due to the small sample size, we were not able to subdivide the groups any further as the results would bear no statistical significance.

### 2.2. Settings, Procedure & Data Collection

To ensure accurate recall of information, all participants were interviewed less than two months after being discharged from the psychiatric ward. The data were recorded in their medical records, which are stored in the military’s Computerized Patient Record system. Our study included only the IDF soldiers who attempted suicide between March 2014 and February 2016. During this time frame about thirty soldiers died by suicide in the IDF [33]. Study participation was limited to those who required at least a 24 h stay at the hospital following the attempt. Over this period, 341 IDF soldiers met these criteria [32]. Out of this group our investigators were able to reach 170 soldiers, since many were discharged from service immediately and did not reply to our attempts to make contact. Out of those who did respond, 65 gave their consent to take part in the study. 

All of the questionnaires (see below), the SBQ-R, SHBQ, C-SSRS, and BSS, were administered to the participants in Hebrew. The translation was performed by a professional translator, to ensure the retained sensitivity and specificity of the measures. In addition, all of the interviews were administered in person by the same IDF Mental Health professional to prevent inconsistency between interviewers.

### 2.3. Measures

#### 2.3.1. Demographic and Personal Variables

The soldiers’ demographics included their gender, age, type of service (Combat/Non-Combat), months served prior to most recent suicide attempt (0–6; 7–12; >12 months), and Intelligence Rating Score (IRS; a cognitive test measuring an aspect of intellectual ability, it is equivalent to a normally distributed IQ score) [34].

#### 2.3.2. Psychiatric Diagnosis

Since known mental illness increases the risk for suicide, diagnosed psychiatric illnesses were included [14,35,36] via the ICD-10. Psychiatric diagnosis before and after the suicide attempt was used as a dichotomous variable (No/Yes for anxiety and mood disorders and personality disorders; see Table 1).

#### 2.3.3. Suicidal Behavior Variables

The study utilized four standard measures of suicidal ideation and behavior: Suicidal Behaviors Questionnaire-Revised (SBQ-R) [37], Self-Harm Behavior Questionnaire (SHBQ) [38], Columbia Suicide Severity Rating Scale (C-SSRS) [39,40], and Beck Scale for Suicide Ideation (BSS) [41]. The characteristics of the suicide attempt were based on C-SSRS and SHBQ scores and include the number of suicide attempts (0; 1; >2), method of first suicide attempt (non-violent (i.e., medications; cutting with only minor bruises or sprains) vs. violent (shooting; immolation; drowning; jumping; hanging); Mann and Malone, 1997) [42], intent to die (Yes/No), and a suicide plan (Yes/No) [32]. The SHBQ was used to determine the total number of self-harm episodes (i.e., non-suicidal self-injury) and was categorized as 0; 1; >2 [32,38].

Suicidal Behaviors Questionnaire-Revised (SBQ-R) [37,38], is a four-item measure that assesses lifetime suicide ideation and attempts, frequency of suicidal ideation during the past year, the extent to which individuals have disclosed suicide thoughts/plans to others, and the likelihood of engaging in future suicidal behavior. The total scores range from 3 to 18, with scores greater than or equal to 7 indicating elevated risk of suicide in the adult general population and a score of 8 or higher indicating elevated risk in clinical settings. This questionnaire has been shown to have strong psychometric properties (e.g., α = 0.82–0.88) [37]. For more details on SBQ-R analyses by this study group see the reference [32].

Self-Harm Behavior Questionnaire (SHBQ) [38], is a 27-item four-part interview designed to assess a range of self-harm behaviors. The first section examines self-harm behavior (SH) that is not suicidal in nature (i.e., non-suicidal self-injury). The second section examines suicide attempts (SA), the third suicide threats (ST), and the fourth suicide ideation (SI). Items include a combination of binary and Likert-type questions regarding these four suicide-relevant domains. Total scores for each subscale are calculated with higher scores indicating that the individual endorsed more items in that domain [32,38]. The SHBQ was utilized to determine the age at which initial suicidal ideation was experienced (<18 or ≥18 years old) and the age at which the most recent suicidal ideation was experienced (<14 years; 15–17 years; ≥18 years old). Ideation is defined as internal thoughts not verbally expressed to anyone. It should be noted that some participants only experienced one episode of suicidal ideation, thus their most recent episode is also their first episode of suicidal ideation. The SHBQ was also used to determine the age at which the most recent suicidal threats (ST) were expressed (<14; 15–17; ≥18 years old) and the age at which the initial (also known as first lifetime) suicidal threats were made (<14; 15–17; ≥18 years old). A threat is defined as a verbal expression of intention to someone else. The SHBQ has displayed strong psychometric properties (subscale α’s = 0.88–0.96) in previous studies [38]. In the current study Cronbach’s α was as follows: SH—α = 0.93; SA—α = 0.81; ST—α = 0.92; and SI—α = 0.73 (for more details of the SHBQ analyses by this study group see the reference [32]). The Columbia Suicide Severity Rating Scale (C-SSRS) [39], was administered as a semi-structured interview. Our study utilized two components: potential severity and actual severity of the most recent suicide attempt. This scale quantifies both suicidal ideation and behavior, with a higher score indicating greater severity. Reliability and validity of this scale has been documented [39,40]. The first five items on the scale refer to suicide ideation with binary (Yes/No) answer choices. The next five items assess the intensity of suicidal ideation on a scale of 0 (suicidal ideation denied; i.e., no suicidal ideation) to 5 (suicidal ideation with a plan; i.e., severe suicidal ideation). The other items assess frequency and duration, along with deterrents and motivators of ideation. The total score reflects only the items with a score greater than 0 and thus the totals range between 1 and 25. Cronbach’s alpha coefficient has been reported as α = 0.80 for ideation severity and α = 0.67 for intensity when used with IDF soldiers [32].

Beck Scale for Suicide Ideation (BSS; Beck) [41] is a 19-item self-report questionnaire designed to assess suicidal thoughts, plans, and intent to die by suicide. The first five items are screening items, while the last two items assess intent and previous attempts. Each item’s score ranges from 0 to 2, resulting in an overall score of 0 to 38. The BSS has strong psychometric properties. Cronbach’s alpha coefficient α = 0.90 was previously reported [43].

### 2.4. Data Analysis

Analyses were conducted using SPSS Statistics (Chicago, IL, USA) Version 23.0 (IBM, 2015). Descriptive statistics were performed followed by univariate Chi-square or Fishers’ Exact Test as appropriate. In addition, ANOVA testing was performed on the BSS as a continuous variable with a significance level of *p* = 0.05. Significant results from the univariate analyses were used for further multivariate analyses using General Linear Model (GLM) with binary logistic dependent variables: “potential severity of last attempt” and “actual severity of damage from last suicide attempt”.

### 2.5. Ethical Approval

The Institutional Review Board of the IDF Medical Corps approved the study and waived the requirement for informed consent on the basis of preserving participant anonymity.

## 3. Results

Table 1 presents the demographic and clinical characteristics of the study subjects, categorized by SA before enlistment (*n* = 25) and First SA in service groups (*n* = 40). Significant findings included a higher proportion of females in the SA before enlistment group (56.0%), compared to a higher proportion of males in the First SA in service group (72.5% *p* = 0.035). In addition, only 4% of the SA before enlistment group were combat soldiers, compared with 25% in the First SA in service group (*p* = 0.04). Both groups had similar characteristics with regard to other variables.

Table 2 shows the comparison of characteristics of suicidal thoughts and behaviors between the SA before enlistment and First SA in service groups. In the first group, the median age for the initial lifetime episode of Suicidal Ideation was 15.0 years old, whereas in the second group, the median age was 17.0 years old (*p* = 0.03). Most individuals in the SA before enlistment group reported that their first experience of suicidal ideation was either prior to the age of 14 (41.2%; *n* = 7) or at 15–17 years old (41.2%; *n* = 7). This was in marked contrast to the majority of the First SA in service group, which reported that they did not have suicidal thoughts before the age of 18 (73.9%; *n* = 17, *p* = 0.001). In addition, 80% of the soldiers in the SA before enlistment group had moderate or severe BSS scores, compared to 60% of the First SA in service group (*p* = 0.027).

For most of the subjects in the First SA in service group, the current suicide attempt was the only attempt they had ever made, but for almost 30% (29.7%; *n* = 11) at least one more suicide attempt had been made during their military service. 

Finally, a multivariate analysis was performed to predict the potential (Table 3) and actual severity (Table 4) of suicide attempts. This analysis showed that having the most recent episode of suicidal ideation at the age of 18 or older (OR = 1.815; 95% CI, 1.151–2.863; *p* = 0.010) and having a single suicide attempt (OR = 2.760; 95% CI, 1.318–5.779; *p* = 0.007) during military service were the most significant predictors for serious potential lethality (i.e., placed a loaded gun in his/her mouth and pulled the trigger but the gun failed to fire, thus producing no medical damage). Being a non-combat soldier serves as the most significant risk factor of the actual severity (defined as degree of physical, medical damage) caused by the attempt (OR = 2.88; 95% CI, 0.127–0.653; *p* = 0.011).

## 4. Discussion

Serious suicide attempts during military service, that fortunately do not end in death, warrant investigation and understanding, as they serve as a window into the minds of those who perish by suicide [44]. This knowledge is even more critical when the first suicide attempt that occurs during military service is severe, sudden, and without a previous history of suicidal behavior [45]. In a recent study it was found that among soldiers with no previous suicide attempts, emotion relief motives were associated with significantly increased risk for suicide attempt during the 6-month follow-up assessment. Among soldiers with a prior attempt, emotion relief motives were not associated with later suicide attempts [46]. Better understanding of subgroup differences in soldiers who make suicide attempts may lead to improved assessment and management approaches.

Our aim in the current study was to investigate the differences in suicide attempt characteristics of service members comparing those who reported suicidal behavior before enlisting with those who made their first suicide attempt after enlisting. Within those subgroups, suicidal lethality and intent was then examined, in order to appreciate the impact of past behavior on current attempt.

The full suicidal history of the subjects in this study was not known, in most cases, before the last suicide attempt, but rather came to light only during the hospitalizations and study interviews. Screening the medical history of military recruits prior to enlistment based on self-report is quite a challenge [45]. In the course of our investigation, we found that about half of the study population had prior suicidal behavior and concealed it from the military authorities, putting themselves at further risk for suicidal behavior during the stressful time of service. The finding that the younger the age of first suicide attempt, the more future attempts are made is not surprising, given previous research on civilians and soldiers [5,47,48]. The finding that soldiers who made two or more attempts also reported prior suicidal behavior is also consistent with the existing literature [49,50].

The most significant finding in the current study is that recent suicidal ideation, rather than past suicide attempts, is the strongest predictor for severe suicide attempts. This finding is in concordance with recent studies which emphasized the significance of suicidal ideation among soldiers who recently enlisted and the risk of them transitioning to suicidal acts [45,46]. 

Moreover, no significant differences in the severity of the suicide attempts (either actual or potential severity) were found between those who had prior suicide attempts and those who made their first attempt while in the service. As a matter of fact, most of the suicide attempts that occurred in service for the first time used a violent method and therefore had high potential lethality. Perhaps, in retrospect, the findings are not surprising, since we selected only the soldiers whose suicide attempts were defined by the treating psychiatrists as serious. Our results suggest that the presence of severe suicidal ideation should therefore concern clinicians, even if there is no known previous history of suicidal behavior. For a variety of reasons related to the nature of military service, soldiers may prefer to share information about suicide ideation in anonymous surveys, instead of reporting their suicidal thoughts during a suicide assessment conducted by a clinician [51].

Previous research has shown that suicidality in soldiers with no prior history of suicidal behavior might be explained by impulsive traits [52,53]. Therefore, the current findings are consistent with our previous studies where impulsivity made suicide attempts less predictable [19]. Psychological autopsies of soldiers who killed themselves in the Israeli military indicate that about half of all suicides were impulsive and without previous suicidal behavior or signs. Further supporting the possible impulsive nature of the attempts is the mild severity of suicidal thoughts (measured by the BSS scale) in the First SA in service group, compared to the severe nature of suicidal thoughts in the soldiers who attempted suicide in the SA before enlistment group. 

Being a non-combat soldier was found to be a significant risk factor of the actual severity of the attempt. This is probably due to the enlistment process in the IDF, allowing the induction of adolescents with known psychiatric disorders to non-combat service [21,22]. These findings are supported by those of a recent study conducted in the IDF [33] which revealed that the risk of suicide amongst non-combat soldiers is much higher than amongst combat soldiers.

Lastly, we found that while females are more likely to attempt suicide before enlistment, males are more likely to make a first attempt during their service. This finding could be the result of the different roles assigned to soldiers in the military, with males being more often assigned to the more stressful divisions (combat and combat-support). Firearms are also less accessible to females, since roles requiring weapons access are mainly performed by male soldiers. This finding indicates that different roles within the military can impact the demographics of suicidal behavior [54]. 

Military service in Israel is compulsory for men and women from the age of 18. During the enlistment process, those suspected of suffering from mental illness are referred to psychological and psychiatric evaluations. Specific procedures and treatment management are tailored in order to support them during their military service [22,33]. Severe suicide attempts during military service are as likely to occur in those who have a history of suicidal behaviors as in those who deny such past behavior. Therefore, current suicidal ideation and behavior along with male gender, a tendency to impulsive behavior, and previously diagnosed psychiatric disorders are critical in risk assessment for suicidality in this population. This study emphasizes the need for a deeper understanding of impulsive personality traits and the relationship between impulsiveness, suicide ideation, and severe suicidal behavior.

### Limitations

This study has several limitations. The conclusions are constrained by the relatively small sample size and the use of a convenience sample. In addition, the G power 3 statistical power analysis software indicated a required sample size of *n* = 84 to obtain statistical power = 0.80 for correlational analyses, assuming a medium effect size of *r* = 0.30. Therefore, due to our restricted inclusion criteria we were slightly under-powered. Another limitation is the reliability of the subject’s responses. This may be influenced by the soldier’s desire to remain or leave military service following an attempt. The data were also obtained using retrospective self-report measures which can introduce biases caused by factors such as mood-dependent recall, failure to recall information, and social desirability. However, previous research supported the psychometric properties of the tools used in this study with soldiers making serious suicide attempts [32].

Another important aspect that was not covered in this study is the possible motivation for enlistment of soldiers with prior history of suicidality, such as a means to realize suicidal wishes in the military setting. In other words, some individuals may not care if they are assigned hazardous roles during their military service. Exploring these motivations could be done through clinical interviews and might uncover important information that could influence future screening strategies and interventions. These limitations notwithstanding, the findings from this study highlight the importance of not only screening for at-risk individuals (i.e., prior suicidal behavior) but the need for determining how protective it is to place soldiers in noncombat units since individuals in these units make the most serious suicide attempts during their military service.

## Figures and Tables

**Table 1 ijerph-18-00649-t001:** Suicidal Characteristics by Suicide Attempt Before Enlistment and First SA in service group.

Variables	Parameter	Categories	SA before Enlistment (*n* = 25; 38.5%)		First SA in Service (*n* = 40; 61.5%)		Total		Exact Sig. (2-Sided)
			N	%	N	%	N	%	
**Demographics**	**Gender**	Male	11	44.0	29	72.5	40	61.5	
		Female	14	56.0	11	27.5	25	38.5	0.035
	**Months Served**	0–6	14	60.9	19	50.0	33	54.1	
		7–12	7	30.4	6	15.8	13	21.3	
		>12	2	8.7	13	34.2	15	24.6	0.067
	**Type of Service**	Combat	1	4.0	10	25.0	11	16.9	
		Non-Combat	24	96.0	30	75.0	54	83.1	0.040
	**Intelligence Score**	<100	11	45.8	10	25.6	21	33.3	
		100–118	10	41.7	24	61.5	34	54.0	
		≥118	3	12.5	5	12.8	8	12.7	0.224
**Psychiatric Diagnosis**	**Before Suicide Attempt**	Anxiety & Mood Disorders	14	56.0	15	37.5	29	44.6	
		Personality Disorders	5	20.0	13	32.5	18	27.7	
		No Diagnosis	6	24.0	12	30.0	18	27.7	0.351
	**After Suicide Attempt**	Anxiety and mood disorders	17	68.0	23	57.5	40	61.5	
		Personality Disorder	6	24.0	16	40.0	22	33.9	
		No Diagnosis	2	8.0	1	2.5	3	4.6	0.270

**Table 2 ijerph-18-00649-t002:** Comparison of characteristics of suicidal thoughts and behaviors.

Variables	Parameter	Categories	SA before Enlistment (*n* = 25; 38.5%)		First SA in Service (*n* = 40; 61.5%)		Total		*p*-Value
			N	%	N	%	N	%	
**Suicidal Ideation**	**Age of Initial Suicidal Ideation**	<14	7	41.2	4	17.4	11	27.5	
		15–17	7	41.2	2	8.7	9	22.5	
		≥18	3	17.6	17	73.9	20	50.0	0.001
	**Age of Last Suicidal Ideation ^π^**	<14	7	31.8	5	25.0	12	28.6	
		15–17	11	50.0	4	20.0	15	35.7	
		≥18	4	18.2	11	55.0	15	35.7	0.043
	**Beck Scale for Suicide Ideation**	Mild (0–10)	5	20.0	16	40.0	21	32.3	
		Moderate/Severe (>11)	20	80.0	24	60.0	44	67.7	0.027
	**Age of Most Recent Suicidal Threats**	<14	0	0.0	0	0.0	0	0.0	
		15–17	2	11.8	2	8.7	4	10.0	
		≥18	15	88.2	21	91.3	36	90.0	1.000
**Recent Suicide Attempt**	**Intention to die**	Yes	10	41.7	15	38.5	25	39.7	
		No	14	58.3	24	61.5	38	60.3	1.000
	**Suicide Plan**	Yes	24	96.0	33	86.8	57	90.5	
		No	1	4.0	5	13.2	6	9.5	0.389
	**Method of Suicide attempt**	Non-Violent ^∞^	13	54.2	15	41.7	28	46.7	
		Violent	11	45.8	21	58.3	32	53.3	0.431
	**Potential Severity ^#^**	No injury ^€^	3	13.6	2	5.9	5	8.9	
		Injury but not likely to cause death ^€^	10	45.5	11	32.4	21	37.5	
		Potential death despite available medical care ^€^	9	40.9	21	61.8	30	53.6	0.321
	**Actual Severity ^#^**	No physical damage	10	45.4	15	44.1	25	44.6	
		Minor physical damage	8	36.4	9	26.5	17	30.4	
		Moderate or Severe physical damage	4	18.2	10	29.4	14	25.0	0.574
**Past Non-Suicidal Self-Harm**	**Number of NSSI Episodes**	0	11	44.0	20	54.1	31	50.0	
		1	0	0.0	2	5.4	2	3.2	
		≥2	14	56.0	15	40.5	29	46.8	0.366

Note: ^π^ For 55.0% of the sample (*n* = 11), the age of most recent suicidal ideation and the age of initial suicidal ideation is the same, as their most recent episode of suicidal ideation was also their first; ∞ non-violent (medications or other ingested substances), cutting (minor bruises or sprains), violent (shooting; immolation; drowning; jumping; hanging); ^#^ of most recent suicide attempt; ^€^ behavior not likely to result in injury. Potential severity and actual severity of the most recent suicide attempt quantifies both suicidal ideation and behavior, with a higher score indicating greater severity. The first five items on the scale refer to suicide ideation with binary Yes/No answer choices. The next five items assess the intensity of suicidal ideation on a scale of 0 (suicidal ideation denied; i.e., no suicidal ideation) to 5 (suicidal ideation with a plan; i.e., severe suicidal ideation). The other items assess frequency and duration, along with deterrents and motivators of ideation. The total score reflects only the items with a score greater than 0 and thus the totals range between 1 and 25.

**Table 3 ijerph-18-00649-t003:** Multivariate Analysis (GLM)* of Potential Clinical Severity of the Soldiers’ Most Recent Suicide Attempt (N = 65).

Parameter (Intercept)	Categories	B	Hypothesis Test	Exp(B)	95% Wald Confidence Interval for Exp(B)	
			Sig.		Lower	Upper
**Age of Initial Suicidal Ideation**		1.346	0.000	3.843	3.029	4.876
	≥18	0.269	0.201	1.309	0.867	1.977
	<18	0 ^a^		1		
**Age of Last known Suicidal Ideation**		1.250	0.000	3.490	2.513	4.848
	≥18	0.596	0.010	1.815	1.151	2.863
	15–17	0.107	0.639	1.113	0.711	1.742
	≤14	0 ^a^		1		
**Age of Last threats**		1.400	0.000	4.055	2.656	6.190
	≥18	0.163	0.555	1.176	0.686	2.017
	15–17	−0.150	0.643	0.861	0.456	1.623
	≤14	0 ^a^		1		
**Beck Scale for Suicide Ideation**		1.529	0.000	4.615	3.406	6.255
	Mild (0–10)	0.137	0.603	1.147	0.684	1.923
	Moderate (11–20)	−0.196	0.312	0.822	0.562	1.203
	Severe (21–38)	0 ^a^		1		
**Gender**		1.391	0.000	4.020	3.081	5.245
	Male	0.094	0.597	1.098	0.777	1.553
	Female	0 ^a^				
**Combat**		1.457	0.000	4.291	3.554	5.181
	Combat	−0.057	0.804	0.945	0.605	1.476
	Non-Combat	0 ^a^				
**Number of Suicide Attempts**		0.667	0.059	1.948	0.974	3.896
	0	0.713	0.055	2.039	0.985	4.224
	1	1.015	0.007	2.760	1.318	5.779
	2	0 ^a^		1		
**Number of NSSI Episodes**		1.400	0.000	4.055	3.136	5.244
	0	0.081	0.655	1.085	0.759	1.550
	1	0.100	0.836	1.105	0.430	2.843
	≥2	0 ^a^		1		

Note: * Multivariate analyses using General Linear Model (GLM) with binary logistic dependent variable of potential clinical severity of the soldiers’ most recent suicide attempt (N = 65); ^a^ Set to zero because this parameter is redundant.

**Table 4 ijerph-18-00649-t004:** Multivariate Analysis (GLM)* of Actual Clinical Severity of the Soldiers’ Most Recent Suicide Attempt (N = 65).

Parameter (Intercept)	Categories	B	Hypothesis Test	Exp(B)	95% Wald Confidence Interval for Exp(B)	
			Sig.		Lower	Upper
**Age of Initial Suicidal Ideation**		0.840	0.001	2.316	1.415	3.793
	≥18	0.468	0.277	1.596	0.687	3.709
	<18	0 ^a^		1		
**Age of Last known Suicidal Ideation ^π^**		0.455	0.219	1.575	0.763	3.253
	≥18	0.853	0.090	2.347	0.876	6.286
	15–17	0.688	0.164	1.990	0.755	5.245
	≤14	0 ^a^		1		
**Age of Last threats**		0.889	0.025	2.432	1.121	5.279
	≥18	0.346	0.479	1.414	0.542	3.686
	15–17	−0.514	0.373	0.598	0.193	1.851
	≤14	0 ^a^		1		
**Beck Scale for Suicide Ideation**		1.000	0.000	2.718	1.585	4.661
	Mild (0–10)	0.750	0.131	2.117	0.801	5.596
	Moderate (11–20)	−0.200	0.565	0.819	0.414	1.619
	Severe (21–38)	0 ^a^		1		
**Gender**		1.167	0.000	3.211	1.986	5.193
	Male	−0.292	0.369	0.747	0.396	1.411
	Female	0 ^a^				
**Combat**		1.170	0.000	3.223	2.322	4.472
	Non-Combat	1.059	0.011	2.88	0.127	0.653
	Combat	0 ^a^				
**Number of Suicide Attempts**		1.250	0.040	3.490	1.057	11.524
	0	−0.250	0.701	0.779	0.218	2.785
	1	−0.295	0.656	0.744	0.203	2.727
	2	0 ^a^		1		
**Number of NSSI Episodes**		1.111	0.000	3.038	1.947	4.740
	2	−0.265	0.414	0.767	0.406	1.448
	1	−0.611	0.480	0.543	0.100	2.955
	0	0 ^a^				

Note: * Multivariate analyses using General Linear Model (GLM) with binary logistic dependent variable of actual clinical severity of the soldiers’ most recent suicide attempt (N = 65); The dependent variable is the potential severity of the last suicide attempt. ^π^ age of last known suicidal ideation (≤14, *n* = 12 (28.6%); ≤17, *n* = 15 (35.7%); ≥18, *n* = 15 (35%); For 35.0% of the sample (*n* = 15), the age of last suicidal ideation is the same as the age of initial suicidal ideation, as their last episode of suicidal ideation was also their first episode; ^a^ Set to zero because this parameter is redundant.

## Data Availability

Due to IDF rules regarding patient medical records, a primary source of data for this study, no study data have been placed in a public repository. Questions about data can be directed to the first author at lshelef4@gmail.com.

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
