# Peer review of "The Role of Past Suicidal Behavior on Current Suicidality: A Retrospective Study in the Israeli Military"

_ijerph, 2021, doi:10.3390/ijerph18020649_

Round 1
Reviewer 1 Report
Here Shelef et al present an interesting study on suicide in a specific population, as suicide in a military setting is of special interest, the relevance of the study is not minor. However, some weaknesses should be acknowledged: first, and recognized by authors as a limitation is the small sample size of just 65 people. Second, and more important, I have concerns about how the study is presented, whereas the focus of the study is in fact the severity (or lethality) of suicide attempt authors do not explain this in their aim, and the study aim is presented very generally. This major limitation must be overcome and the manuscript should be rewritten clarifying the study focus, consequently, the authors must change their aim and write their introduction and discussion keeping this aim in mind.
Next, I detail other aspects regarding the manuscript
ABSTRACT: Results must be presented in a logical order, from univariant analysis to multivariate analyses
INTRODUCTION:
My main concern is how authors present their aim, they write it in a general way instead to explain that their interest is the severity of suicide attempt. I would present the aim by explaining that they want to study differences between soldiers who commit their first SA during military service against those with previous SA and after that specifically studying factors influencing the severity of SA
Accordingly with these changes, the introduction must be re-written and review factors underlying serious suicide attempts
METHODS: This section is overall well written and explained, nevertheless I would acknowledge knowing why only 65 soldiers out of 341 who made a suicide attempt were included in the study with a more detailed explanation more than "only 65 of these soldiers met the inclusion 121 criteria and agreed to take part in the study" (lines 120-122). Also, in measures 2.3.2 (lines 134 onwards) although a dichotomous category is interesting for logistic regression the detailed diagnoses must be listed and afterwards shown in Results
Finally, I would consider moving Table 1 to the Results section
RESULTS:
First of all, and given that most authors consider that violent suicide attempts are close to death by suicide (see as an example 10.4088/JCP.13m08524) it would be interesting to have information about death by suicide within the study time period
Table 1 and its explanation in text: Due to the dichotomization data from Table 1 regarding diagnoses and text information (lines 94-95) are different
Table 1 and Table 2 are not consistent regarding the naming of the two groups, and in both tables, they are different than names proposed in the introduction: SA before enlistment and First SA in service
In the multivariate analyses the outcomes variables (potential and actual severity should be explained. Furthermore, authors must explain the selection of predictors variables
DISCUSSION:
My main concern is that the discussion is excessively short and the authors do not discuss in depth their results; furthermore, they do not put sufficiently in context with previous research. Discussion should be improved
Also, more limitations than the two describe should be acknowledged as the truthfulness of the information regarding psychiatric issues provided by this specific population or the lack of other variables related to the severity of suicide attempts
Author Response
We would like to thank reviewer #1 for his comments as well as for finding our current study interest and relevant.

Reviewer 2 Report
The manuscript “The Role of Past Suicidal Behavior on Current Suicidality: A Retrospective Study in the Israeli Military” is a novel article of great relevance that studies suicidal behavior in a risk group such as the military services.
Here are some suggestions for improvement in the article:
- Line 25. “n”
- Line 115. Was the sample lost solely because it did not meet the inclusion criteria? What percentage of participants refused to participate?
- Was information obtained on other types of variables such as whether they had been in treatment after the attempt or whether they were currently taking medication?
- Line 147. Typing error.
- Add in the section of questionnaires the reliability obtained in previous studies in the first two tests commented
- Line 172. The Columbia Suicide Severity Rating Scale
- Line 209. Typing error (p)
- You may want to point out the word "intercept" at the beginning of tables 3 and 4 so as not to have to repeat it throughout the table.
- Add reference for the affirmation of line 250
- Adjust reference style to the format required in the MDPI journals https://www.mdpi.com/authors/references
- It would be interesting to add future lines of research on the importance of detecting early and working on impulsivity or preventing future attempts in a suitable way among this group.
Author Response
Reviewer comment - The manuscript “The Role of Past Suicidal Behavior on Current Suicidality: A Retrospective Study in the Israeli Military” is a novel article of great relevance that studies suicidal behavior in a risk group such as the military services.
Reply: We would like to thank reviewer #2 for his comments as well as for finding our current study a novel article of great relevance that studies suicidal behavior in a risk group such as the military services. A co-author who is a native English speaker/writer has carefully reviewed the manuscript for correct language use and edited accordingly.
Here are some suggestions for improvement in the article:
- Line 25. “n”.
Reply: We correct the n. line 25
- Line 115. Was the sample lost solely because it did not meet the inclusion criteria? What percentage of participants refused to participate?
Reply: We accept the comments and add explanation, as follow:
Line 153-156. Over this period, 341 IDF soldiers met these criteria (Chiurliza et al., 2018). Out of this group our investigators were able to reach 170 soldiers, since many were discharged from service immediately and did not reply to our attempts to make contact. Out of those who did respond, 65 gave their consent to take part in the study.
- Was information obtained on other types of variables such as whether they had been in treatment after the attempt or whether they were currently taking medication?
Reply: We accept the comments and add explanation, as follow:
Lines 114-116. Under military procedures and following regulations of chain of care, those who remained in military service, continued to be treated and observed by military mental health professionals throughout their service.
- Line 147. Typing error.
Reply: We correct the typing error. Line 184
- Add in the section of questionnaires the reliability obtained in previous studies in the first two tests commented
Reply: We accept the comments and add the reliability and explanation, as follow:
Suicidal Behaviors Questionnaire-Revised SBQ-R. Lines 193-195. This questionnaire has been shown to have strong psychometric properties (e.g., α = .82–.88; Osman et al., 2001). For more details on SBQ–R analyses by this study group see Chiurliza et al., 2018.
Self-Harm Behavior Questionnaire SHBQ. Lines 210-213. The SHBQ has displayed strong psychometric properties (subscale α’s = .88–.96) in previous studies (Gutierrez et al., 2001). In the current study Cronbach’s α was as follows: SH - α = .93; SA - α = .81; ST - α = .92; and SI - α = .73 (for more details of the SHBQ analyses by this study group see Chiurliza et al., 2018).
- Line 172. The Columbia Suicide Severity Rating Scale
Reply: We correct to the Italic font. Line 214
- Line 209. Typing error (p)
Reply: We correct the typing error. Line 253
- You may want to point out the word "intercept" at the beginning of tables 3 and 4 so as not to have to repeat it throughout the table.
Reply: We made this change appropriately in Tables 3 and 4.
- Add reference for the affirmation of line 250
Reply: We correct and add references.
Lines 327-331 The most significant finding in the current study is that recent suicidal ideation, rather than past suicide attempts, is the strongest predictor for severe suicide attempts. This finding is in concordance with recent studies which emphasized the significance of suicidal ideation among soldiers who recently enlisted and the risk for them transitioning to suicidal acts (Bryan, May, & Harris, 2019; Smith et al., 2020).
- Adjust reference style to the format required in the MDPI journals https://www.mdpi.com/authors/references
Reply: We made the changes appropriately.
- It would be interesting to add future lines of research on the importance of detecting early and working on impulsivity or preventing future attempts in a suitable way among this group.
Reply: We accept the comments and add as follows,
Lines 379-381. This study emphasizes the need for a deeper understanding of impulsive personality traits and the relationship between impulsiveness, suicide ideation, and severe suicide behavior.

Reviewer 3 Report
The article is really very interesting and covers an important issue. It would be important to deepen and clarify some aspects, because for the reader some things do not appear immediately very clear.
In the "Introduction" there should be a better description of what enlistment in Israel is like and what kind of evaluations are done to determine eligibility for military service.
In the description of the participants, it should be clarified whether enlistment was voluntary or mandatory.
In the "Discussions" it should be problematized the argument that, as the literature has now widely confirmed, the risk of suicide among military personnel and people who have firearms is higher than the rest of the population, moreover that even those who have already attempted suicide or have suicidal ideations are at greater risk of committing suicide. Certainly something must be changed in the selection strategies for enlistment. In the enrollment phase it would be better not to enroll those who have already attempted suicide or have suicidal ideation. Why is this selection not taking place?
In the "Discussions", with respect to the result, it would be necessary to take up the topic related to the compulsory or voluntary enlistment. In fact, in the group of participants "SA before enlistment" the voluntary choice of arms could correspond to a suicidal project. Conversely, the "First SA in service" of those forced into enlistment may have staged the suicide attempt to gain exemption.
The "Discussions" should highlight the characteristics that differentiate males and females with particular attention to enlistment type.
In the "Research Limitations" it should be indicated that it has not been verified with SA before enlistment whether they pursued a military career to achieve their suicidal intent. The mixed method, using clinical interview analysis, could have provided a better understanding of why the results were obtained. Regarding the importance of using qualitative methods in research around suicide we suggest to cite: Testoni, De Vincenzo, Zamperini "The words to say it - Qualitative suicide research", in Kolves, Sisask, Varnik, De Leo (eds.), Advancing suicide research, 2020, pp. 129-150, New York: Hogrefe.
In future research it would be interesting to investigate the question of how many "SA before enlistment" died in war actions.
Author Response
Comments and Suggestions for Authors
The article is really very interesting and covers an important issue. It would be important to deepen and clarify some aspects, because for the reader some things do not appear immediately very clear.
Reply: We would like to thank reviewer #3 for his comments, as well as for finding our current study covers an important issue. A co-author who is a native English speaker/writer has carefully reviewed the manuscript for correct language use and edited accordingly.
Reviewer comment - In the "Introduction" there should be a better description of what enlistment in Israel is like and what kind of evaluations are done to determine eligibility for military service.
Reply: We accept the comments and add as follows,
Lines 69-76. In the enlistment process, candidates are interviewed and assessed. Since military service in Israel is compulsory, candidates with emotional or behavioral disturbances referred to a military clinical mental health professional, are classified in two broad categories of diagnosis. Adjustment difficulties defined as a cluster of personality traits limiting functionality and adaptability in the military service context. And psychiatric diagnosis, based on ICD-10 classifications, requiring an assessment of the impact of the classification on the soldier's potential functioning (for more details on the method of diagnosis in the IDF see Shelef et al., 2019a; Shelef et al., 2020).
Reviewer comment - In the description of the participants, it should be clarified whether enlistment was voluntary or mandatory.
Reply: We accept the comments and add as follows,
Line 109-114. 2.1. Participants
The study sample consisted of 65 active duty IDF soldiers (61.5% Male), between 18 and 28 years old (M = 20.4, SD ± 1.3) that agreed to participate in this study, while serving their compulsory military service.
Reviewer comment - In the "Discussions" it should be problematized the argument that, as the literature has now widely confirmed, the risk of suicide among military personnel and people who have firearms is higher than the rest of the population, moreover that even those who have already attempted suicide or have suicidal ideations are at greater risk of committing suicide. Certainly something must be changed in the selection strategies for enlistment. In the enrollment phase it would be better not to enroll those who have already attempted suicide or have suicidal ideation. Why is this selection not taking place?
Reply: We accept this comment and add the following paragraph:
Lines:371-374:
Military service in Israel is compulsory for men and women from the age of 18. During the enlistment process those suspected of suffering from mental illness are referred to psychological and psychiatric evaluations. Specific procedures and treatment management are tailored in order to support them during their military service (Shelef, et al., 2019b; Shelef, et al., 2020).
Reviewer comment - In the "Discussions", with respect to the result, it would be necessary to take up the topic related to the compulsory or voluntary enlistment. In fact, in the group of participants "SA before enlistment" the voluntary choice of arms could correspond to a suicidal project. Conversely, the "First SA in service" of those forced into enlistment may have staged the suicide attempt to gain exemption.
The "Discussions" should highlight the characteristics that differentiate males and females with particular attention to enlistment type.
Reply: We accept this comment and add the following paragraphs:
Lines 109-111. The study sample consisted of 65 active duty IDF soldiers (61.5% Male), between 18 and 28 years old (M = 20.4, SD ± 1.3) that agreed to participate in this study, while serving their compulsory military service.
Line 388-394. Another limitation is the reliability of the subject’s responses. This may be influenced by the soldier’s desire to remain or leave military service following an attempt. The data were also obtained using retrospective self-report measures which can introduce biases caused by factors such as mood-dependent recall, failure to recall information, and social desirability. However, previous research supported the psychometric properties of the tools used in this study with soldiers making serious suicide attempts (Chiurliza et al., 2018).
Reviewer comment - In the "Research Limitations" it should be indicated that it has not been verified with SA before enlistment whether they pursued a military career to achieve their suicidal intent. The mixed method, using clinical interview analysis, could have provided a better understanding of why the results were obtained. Regarding the importance of using qualitative methods in research around suicide we suggest to cite: Testoni, De Vincenzo, Zamperini "The words to say it - Qualitative suicide research", in Kolves, Sisask, Varnik, De Leo (eds.), Advancing suicide research, 2020, pp. 129-150, New York: Hogrefe.
In future research it would be interesting to investigate the question of how many "SA before enlistment" died in war actions.
Reply: We accept the comments and add as follows
Lines 395-400. Another important aspect that was not covered in this study is the possible motivation for enlistment of soldiers with prior history of suicidality, such as a means to realize suicidal wishes in the military setting. In other words, some individuals may not care if they are assigned hazardous roles during their military service. Exploring these motivations could be done through clinical interviews and might uncover important information that could influence future screening strategies and interventions.

Round 2
Reviewer 1 Report
Thank you for cover all my concerns regarding your manuscript. I consider it now ready to be published